# Regularized Nonlinear Acceleration

**Damien Scieur**
INRIA & D.I., UMR 8548,
École Normale Supérieure, Paris, France.
`damien.scieur@inria.fr`

**Alexandre d'Aspremont**
CNRS & D.I., UMR 8548,
École Normale Supérieure, Paris, France.
`aspremon@di.ens.fr`

**Francis Bach**
INRIA & D.I., UMR 8548,
École Normale Supérieure, Paris, France.
`francis.bach@inria.fr`

## Abstract

We describe a convergence acceleration technique for generic optimization problems. Our scheme computes estimates of the optimum from a nonlinear average of the iterates produced by any optimization method. The weights in this average are computed via a simple and small linear system, whose solution can be updated online. This acceleration scheme runs in parallel to the base algorithm, providing improved estimates of the solution on the fly, while the original optimization method is running. Numerical experiments are detailed on classical classification problems.

## 1  Introduction

Suppose we want to solve the following optimization problem

$$\min_{x \in \mathbb{R}^n} f(x) \tag{1}$$

in the variable $x \in \mathbb{R}^n$, where $f(x)$ is strongly convex with respect to the Euclidean norm with parameter $\mu$, and has a Lipschitz continuous gradient with parameter $L$ with respect to the same norm. This class of function is often encountered, for example in regression where $f(x)$ is of the form

$$f(x) = \mathcal{L}(x) + \Omega(x),$$

where $\mathcal{L}(x)$ is a smooth convex loss function and $\Omega(x)$ is a smooth strongly convex penalty function.

Assume we solve this problem using an iterative algorithm of the form

$$x_{i+1} = g(x_i), \quad \text{for } i = 1, ..., k, \tag{2}$$

where $x_i \in \mathbb{R}^n$ and $k$ the number of iterations. Here, we will focus on the problem of estimating the solution to (1) by tracking only the sequence of iterates $x_i$ produced by an optimization algorithm. This will lead to an acceleration of the speed of convergence, since we will be able to extrapolate more accurate solutions without any calls to the oracle $g(x)$.

Since the publication of Nesterov's optimal first-order smooth convex minimization algorithm [1], a significant effort has been focused on either providing more explicit interpretable views on current acceleration techniques, or on replicating these complexity gains using different, more intuitive schemes. Early efforts were focused on directly extending the original acceleration result in [1] to broader function classes [2], allow for generic metrics, line searches or simpler proofs [5, 6], produce adaptive accelerated algorithms [7], etc. More recently however, several authors [8, 9] have started

using classical results from control theory to obtain numerical bounds on convergence rates that match the optimal rates. Others have studied the second order ODEs obtained as the limit for small step sizes of classical accelerated schemes, to better understand their convergence [10, 11]. Finally, recent results have also shown how to wrap classical algorithms in an outer optimization loop, to accelerate convergence [12] and reach optimal complexity bounds.

Here, we take a significantly different approach to convergence acceleration stemming from classical results in numerical analysis. We use the iterates produced by any (converging) optimization algorithm, and estimate the solution directly from this sequence, assuming only some regularity conditions on the function to minimize. Our scheme is based on the idea behind Aitken's $\Delta^2$ algorithm [13], generalized as the Shanks transform [14], whose recursive formulation is known as the $\varepsilon$-algorithm [15] (see e.g. [16, 17] for a survey). In a nutshell, these methods fit geometrical models to linearly converging sequences, then extrapolate their limit from the fitted model.

In a sense, this approach is more statistical in nature. It assumes an approximately linear model holds for iterations near the optimum, and estimates this model using the iterates. In fact, Wynn's algorithm [15] is directly connected to the Levinson-Durbin algorithm [18, 19] used to solve Toeplitz systems recursively and fit autoregressive models (the Shanks transform solves Hankel systems, but this is essentially the same problem [20]). The key difference here is that estimating the autocovariance operator is not required, as we only focus on the limit. Moreover, the method presents strong links with the conjugate gradient when applied to unconstrained quadratic optimization.

We start from a slightly different formulation of these techniques known as minimal polynomial extrapolation (MPE) [17, 21] which uses the minimal polynomial of the linear operator driving iterations to estimate the optimum by nonlinear averaging (i.e., using weights in the average which are nonlinear functions of the iterates). So far, for all the techniques cited above, no proofs of convergence of these estimates were given in the case where the iterates made the estimation process unstable.

Our contribution here is to add a regularization in order to produce explicit bounds on the distance to optimality by controlling the stability through the regularization parameter, thus explicitly quantifying the acceleration provided by these techniques. We show in several numerical examples that these stabilized estimates often speed up convergence by an order of magnitude. Furthermore this acceleration scheme thus runs in parallel to the original algorithm, providing improved estimates of the solution on the fly, while the original method is progressing.

The paper is organized as follows. In section 2.1 we recall basic results behind MPE for linear iterations and we will introduce in section 2.2 a formulation of the approximate version of MPE and make a link with the conjugate gradient method. Then, in section 2.3, we generalize these results to generic nonlinear iterations and show, in section 2.4, how to fully control the impact of nonlinearity. We use these results to derive explicit bounds on the acceleration performance of our estimates.

## 2  Approximate Minimal Polynomial Extrapolation

In what follows, we recall the key arguments behind *minimal polynomial extrapolation (MPE)* as derived in [22] or also [21]. We also explain a variant called *approximate minimal polynomial extrapolation (AMPE)* which allows to control the number of iterates used in the extrapolation, hence reduces its computational complexity. We begin by a simple description of the method for linear iterations, then extend these results to the generic nonlinear case. Finally, we fully characterize the acceleration factor provided by a regularized version of AMPE, using regularity properties of the function $f(x)$, and the result of a Chebyshev-like, tractable polynomial optimization problem.

### 2.1  Linear Iterations

Here, we assume that the iterative algorithm in (2) is in fact linear, with

$$x_i = A(x_{i-1} - x^*) + x^*, \tag{3}$$

where $A \in \mathbb{R}^{n \times n}$ (not necessarily symmetric) and $x^* \in \mathbb{R}^n$. We assume that 1 is not an eigenvalue of $A$, implying that (3) admits a unique fixed point $x^*$. Moreover, if we assume that $\|A\|_2 < 1$, then $x_k$ converge to $x^*$ at rate $\|x_k - x^*\|_2 \leq \|A\|_2^k \|x_0 - x^*\|$. We now recall the *minimal polynomial extrapolation* (MPE) method as described in [21], starting with the following definition.

**Definition 2.1** *Given $A \in \mathbb{R}^{n \times n}$, s.t. 1 is not an eigenvalue of $A$ and $v \in \mathbb{R}^n$, the minimal polynomial of $A$ with respect to the vector $v$ is the lowest degree polynomial $p(x)$ such that*

$$p(A)v = 0, \qquad p(1) = 1.$$

Note that the degree of $p(x)$ is always less than $n$ and the condition $p(1) = 1$ makes $p$ unique. Notice that because we assumed that 1 is not an eigenvalue of $A$, having $p(1) = 1$ is not restrictive since we can normalize each minimal polynomial with the sum of its coefficients (see Lemma A.1 in the supplementary material). Given an initial iterate $x_0$, MPE starts by forming a matrix $U$ whose columns are the increments $x_{i+1} - x_i$, with

$$u_i = x_{i+1} - x_i = (A - I)(x_i - x^*) = (A - I)A^i(x_0 - x^*). \tag{4}$$

Now, let $p$ be the minimal polynomial of $A$ with respect to the vector $u_0$ (where $p$ has coefficients $c_i$ and degree $d$), and $U = [u_0, u_1, ..., u_d]$. So

$$\sum_{i=0}^{d} c_i u_i = \sum_{i=0}^{d} c_i A^i u_0 = p(A)u_0 = 0 , \qquad p(1) = \sum_{i=0}^{d} c_i = 1. \tag{5}$$

We can thus solve the system $Uc = 0$, $\sum_i c_i = 1$ to find $p$. In this case, the fixed point $x^*$ can be computed *exactly* as follows

$$
\begin{aligned}
0 = \sum_{i=0}^{d} c_i A^i u_0 \quad &= \quad \sum_{i=0}^{d} c_i A^i (A - I)(x_0 - x^*) \\
&= \quad (A - I)\sum_{i=0}^{d} c_i A^i (x_0 - x^*) = (A - I)\sum_{i=0}^{d} c_i (x_i - x^*).
\end{aligned}
$$

Hence, using the fact that 1 is not an eigenvalue of $A$ and $p(1) = 1$,

$$(A - I)\sum_{i=0}^{d} c_i(x_i - x^*) = 0 \quad \Leftrightarrow \quad \sum_{i=0}^{d} c_i(x_i - x^*) = 0 \quad \Leftrightarrow \quad \sum_{i=0}^{d} c_i x_i = x^*.$$

This means that $x^*$ is obtained by *averaging* iterates using the coefficients in $c$. The averaging in this case is called nonlinear, since the coefficients of $c$ vary with the iterates themselves.

## 2.2 Approximate Minimal Polynomial Extrapolation (AMPE)

Suppose now that we only compute a fraction of the iterates $x_i$ used in the MPE procedure. While the number of iterates $k$ might be smaller than the degree of the minimal polynomial of $A$ with respect to $u_0$, we can still try to make the quantity $p_k(A)u_0$ small, where $p_k(x)$ is now a polynomial of degree at most $k$. The corresponding difference matrix $U = [u_0, u_1, ..., u_k] \in \mathbb{R}^{n \times (k+1)}$ is rectangular.

This is also known as the Eddy-Mešina method [3, 4] or reduced rank extrapolation with arbitrary $k$ (see [21, §10]). The objective here is similar to (5), but the system is now overdetermined because $k < \deg(P)$. We will thus choose $c$ to make $\|Uc\|_2 = \|p(A)u_0\|_2$, for some polynomial $p$ such that $p(1) = 1$, as small as possible, which means solving for

$$c^* \triangleq \operatorname{argmin} \; \|Uc\|_2 \quad \text{s.t.} \; \mathbf{1}^T c = 1 \tag{AMPE}$$

in the variable $c \in \mathbb{R}^{k+1}$. The optimal value $\|Uc^*\|_2$ of this problem is decreasing with $k$, satisfies $\|Uc^*\|_2 = 0$ when $k$ is greater than the degree of the minimal polynomial, and controls the approximation error in $x^*$ with equation (4). Setting $u_i = (A - I)(x_i - x^*)$, we have

$$
\begin{aligned}
\| \textstyle\sum_{i=0}^{k} c_i^* x_i - x^* \|_2 \quad &= \quad \|(I - A)^{-1} \textstyle\sum_{i=0}^{k} c_i^* u_i\|_2 \\
&\leq \quad \left\|(I - A)^{-1}\right\|_2 \|Uc^*\|_2.
\end{aligned}
$$

We can get a crude bound on $\|Uc^*\|_2$ from Chebyshev polynomials, using only an assumption on the range of the spectrum of the matrix $A$. Assume $A$ symmetric, $0 \preceq A \preceq \sigma I \prec I$ and $\deg(p) \leq k$. Indeed,

$$\|Uc^*\|_2 = \|p^*(A)u_0\|_2 \leq \|u_0\|_2 \min_{p:p(1)=1} \|p(A)\|_2 \leq \|u_0\|_2 \min_{p:p(1)=1} \max_{A:0 \preceq A \preceq \sigma I} \|p(A)\|_2, \tag{6}$$

where $p^*$ is the polynomial with coefficients $c^*$. Since $A$ is symmetric, we have $A = Q \Lambda Q^T$ where $Q$ is unitary. We can thus simplify the objective function:

$$\max_{A:0 \preceq A \preceq \sigma I} \|p(A)\|_2 = \max_{\Lambda:0 \preceq \Lambda \preceq \sigma I} \|p(\Lambda)\|_2 = \max_{\Lambda:0 \preceq \Lambda \preceq \sigma I} \max_i |p(\lambda_i)| = \max_{\lambda:0 \leq \lambda \leq \sigma} |p(\lambda)|.$$

We thus have

$$\|Uc^*\|_2 \leq \|u_0\|_2 \min_{p:p(1)=1} \max_{\lambda:0\leq\lambda\leq\sigma} |p(\lambda)|.$$

So we must find a polynomial which takes small values in the interval $[0,\sigma]$. However, Chebyshev polynomials are known to be polynomials for which the maximal value in the interval $[0,1]$ is the smallest. Let $C_k$ be the Chebyshev polynomial of degree $k$. By definition, $C_k(x)$ is a monic polynomial[1] which solves

$$C_k(x) = \operatorname*{argmin}_{p:p \text{ is monic}} \max_{x:x\in[-1,1]} |p(x)|.$$

We can thus use a variant of $C_k(x)$ in order to solve the minimax problem

$$\min_{p:p(1)=1} \max_{\lambda:0\leq\lambda\leq\sigma} |p(\lambda)|. \tag{7}$$

The solution of this problem is given in [23] and admits an explicit formulation:

$$\mathcal{T}(x) = \frac{C_k(t(x))}{C_k(t(1))}, \quad t(x) = \frac{2x-\sigma}{\sigma}.$$

Note that $t(x)$ is simply a linear mapping from interval $[0,\sigma]$ to $[-1,1]$. Moreover,

$$\min_{p:p(1)=1} \max_{\lambda:0\leq\lambda\leq\sigma} |p(\lambda)| = \max_{\lambda:0\leq\lambda\leq\sigma} |T_k(\lambda)| = |T_k(\sigma)| = \frac{2\zeta^k}{1+\zeta^{2k}}, \tag{8}$$

where $\zeta$ is

$$\zeta = (1-\sqrt{1-\sigma})/(1+\sqrt{1-\sigma}) < \sigma. \tag{9}$$

Since $\|u_0\|_2 = \|(A-I)(x_0-x^*)\|_2 \leq \|A-I\|_2\|x_0-x^*\|$, we can bound (6) by

$$\|Uc^*\|_2 \leq \|u_0\|_2 \min_{p:p(1)=1} \max_{\lambda:0\leq\lambda\leq\sigma} |p(\lambda)| \leq \|A-I\|_2 \frac{2\zeta^k}{1+\zeta^{2k}} \|x_0-x^*\|_2.$$

This leads to the following proposition.

**Proposition 2.2** *Let $A$ be symmetric, $0 \preceq A \preceq \sigma I \prec I$ and $c_i$ be the solution of* (AMPE). *Then*

$$\left\| \sum_{i=0}^k c_i^* x_i - x^* \right\|_2 \leq \kappa(A-I)\frac{2\zeta^k}{1+\zeta^{2k}}\|x_0-x^*\|_2, \tag{10}$$

*where $\kappa(A-I)$ is the condition number of the matrix $A-I$ and $\zeta$ is defined in* (9).

Note that, when solving quadratic optimization problems, the rate in this bound matches that obtained using the optimal method in [6]. Also, the bound on the rate of convergence of this method is exactly the one of the conjugate gradient with an additional factor $\kappa(A-I)$.

**Remark:**   This method presents a strong link with the conjugate gradient. Denote $\|v\|_B = \sqrt{v^T B v}$ the norm induced by the definite positive matrix $B$. By definition, at the $k$-th iteration, the conjugate gradient computes an approximation $s$ of $x^*$ which follows

$$s = \operatorname{argmin} \|x-x^*\|_A \quad \text{s.t. } x \in \mathcal{K}_k,$$

where $\mathcal{K}_k = \{Ax^*, A^2x^*, ..., A^kx^*\}$ is called a Krylov subspace. Since $x \in \mathcal{K}_k$, we have that $x$ is a linear combination of the element in $\mathcal{K}_k$, so $x = \sum_{i=1}^k c_i A^i x^* = q(A)x^*$, where $q(x)$ is a polynomial of degree $k$ and $q(0) = 0$. So conjugate gradient solves

$$s = \operatorname{argmin}_{q:q(0)=0} \|q(A)x^* - x^*\|_A = \operatorname{argmin}_{\hat{q}:\hat{q}(0)=0} \|\hat{q}(A)x^*\|_A,$$

which is very similar to equation (AMPE). However, while conjugate gradient has access to an oracle which gives the result of the product between matrix $A$ and any vector $v$, the AMPE procedure can only use the iterations produced by (3) (meaning that the AMPE procedure does not need to know $A$). Moreover, we analyze the convergence of AMPE in another norm ($\|\cdot\|_2$ instead of $\|\cdot\|_A$). These two reasons explain why a condition number appears in the rate of convergence of AMPE (10).

## 2.3 Nonlinear Iterations

We now go back to the general case where the iterative algorithm is nonlinear, with

$$\tilde{x}_{i+1} = g(\tilde{x}_i), \quad \text{for } i = 1, ..., k, \tag{11}$$

where $\tilde{x}_i \in \mathbb{R}^n$ and the function $g$ has a symmetric Jacobian at point $x^*$. We also assume that the method has a unique fixed point written $x^*$ and linearize these iterations around $x^*$, to get

$$\tilde{x}_i - x^* = A(\tilde{x}_{i-1} - x^*) + e_i, \tag{12}$$

where $A$ is now the Jacobian matrix (i.e., the first derivative) of $g$ taken at the fixed point $x^*$ and $e_i \in \mathbb{R}^n$ is a second order error term $\|e_i\|_2 = O(\|\tilde{x}_{i-1} - x^*\|_2^2)$. Note that, by construction, the linear and nonlinear models share the same fixed point $x^*$. We write $x_i$ the iterates that would be obtained using the asymptotic linear model (starting at $x_0$)

$$x_i - x^* = A(x_{i-1} - x^*).$$

Running the algorithm described in (11), we thus observe the iterates $\tilde{x}_i$ and build $\tilde{U}$ from their differences. As in (AMPE) we then compute $\tilde{c}$ using matrix $\tilde{U}$ and finally estimate

$$\tilde{x}^* = \sum_{i=0}^{k} \tilde{c}_i \tilde{x}_i.$$

In this case, our estimate for $x^*$ is based on the coefficient $\tilde{c}$, computed using the iterates $\tilde{x}_i$. We will now decompose the error made by the estimation by comparing it with the estimation which comes from the linear model:

$$\left\| \sum_{i=0}^{k} \tilde{c}_i \tilde{x}_i - x^* \right\|_2 \leq \left\| \sum_{i=0}^{k} (\tilde{c}_i - c_i) x_i \right\|_2 + \left\| \sum_{i=0}^{k} \tilde{c}_i (\tilde{x}_i - x_i) \right\|_2 + \left\| \sum_{i=0}^{k} c_i x_i - x^* \right\|_2. \tag{13}$$

This expression shows us that the precision is comparable to the precision of the AMPE process in the linear case (third term) with some perturbation. Also, if $\|e_i\|_2$ is small then $\|x_i - \tilde{x}_i\|_2$ is small as well. But we need more information about $\|c\|_2$ and $\|\tilde{c} - c\|_2$ if we want to go further.

We now show the following proposition computing the perturbation $\Delta c = (\tilde{c}^* - c^*)$ of the optimal solution of (AMPE), $c^*$, induced by $E = \tilde{U} - U$. It will allow us to bound the first term on the right-hand side of (13) (see proof A.2 in the Appendix). For simplicity, we will use $P = \tilde{U}^T \tilde{U} - U^T U$.

**Proposition 2.3** *Let $c^*$ be the optimal solution to* (AMPE)

$$c^* = \underset{\mathbf{1}^T c = 1}{\operatorname{argmin}} \|Uc\|_2$$

*for some matrix $U \in \mathbb{R}^{n \times k}$. Suppose $U$ becomes $\tilde{U} = U + E$ and write $c^* + \Delta c$ the perturbed solution to* (AMPE). *Let $M = \tilde{U}^T \tilde{U}$ and the perturbation matrix $P = \tilde{U}^T \tilde{U} - U^T U$. Then,*

$$\Delta c = - \left( I - \frac{M^{-1} \mathbf{1} \mathbf{1}^T}{\mathbf{1}^T M^{-1} \mathbf{1}} \right) M^{-1} P c^*. \tag{14}$$

We see here that the perturbation can be potentially large. Even if $\|c^*\|_2$ and $\|P\|_2$ can be potentially small, $\|M^{-1}\|_2$ is huge in general. It can be shown that $U^T U$ (the square of a Krylov-like matrix) presents an exponential condition number (see [24]) because the minimal eigenvalue decays very fast. Moreover, the eigenvalues are perturbed by $P$, leading to a potential huge perturbation $\Delta c$, especially if $\|P\|_2$ is comparable (or bigger) to $\lambda_{\min}(U^T U)$.

## 2.4 Regularized AMPE

The condition number of the matrix $U^T U$ in problem (AMPE) can be arbitrary large. Indeed, this condition number is related to the one of Krylov matrices which has been proved in [24] to be exponential in $k$. By consequence, this conditioning problem coupled with nonlinear errors lead to highly unstable solutions $c^*$ (which we observe in our experiments). We thus study a regularized formulation of problem (AMPE), which reads

$$\begin{array}{ll} \text{minimize} & c^T (U^T U + \lambda I) c \\ \text{subject to} & \mathbf{1}^T c = 1 \end{array} \tag{RMPE}$$

The solution of this problem may be computed with a linear system, and the regularization parameter controls the norm of the solution, as shown in the following Lemma (see proof A.3 in Appendix).

**Lemma 2.4** *Let $c_\lambda^*$ be the optimal solution of problem* (RMPE). *Then*

$$c_\lambda^* = \frac{(U^T U + \lambda I)^{-1} \mathbf{1}}{\mathbf{1}^T (U^T U + \lambda I)^{-1} \mathbf{1}} \qquad and \qquad \|c_\lambda^*\|_2 \leq \sqrt{\frac{\lambda + \|U\|_2^2}{k\lambda}}. \tag{15}$$

This allows us to obtain the following corollary extending Proposition 2.3 to the regularized AMPE problem in (RMPE), showing that the perturbation of $c$ is now controlled by the regulaization parameter $\lambda$.

**Corollary 2.5** *Let $c_\lambda^*$, defined in (15), be the solution of problem* (RMPE). *Then the solution of problem* (RMPE) *for the perturbed matrix $\tilde{U} = U + E$ is given by $c_\lambda^* + \Delta c_\lambda$ where*

$$\Delta c_\lambda = -W M_\lambda^{-1} P c_\lambda^* = -M_\lambda^{-1} W^T P c_\lambda^* \qquad and \qquad \|\Delta c_\lambda^*\|_2 \leq \frac{\|P\|_2}{\lambda} \|c_\lambda^*\|_2,$$

*where $M_\lambda = (U^T U + P + \lambda I)$ and $W = \left( I - \frac{M_\lambda^{-1} \mathbf{1} \mathbf{1}^T}{\mathbf{1}^T M_\lambda^{-1} \mathbf{1}} \right)$ is a projector of rank $k - 1$.*

These results lead us to the following simple algorithm.

---

**Algorithm 1** Regularized Approximate Minimal Polynomial Extrapolation (**RMPE**)

---

**Input:** Sequence $\{x_0, x_1, ..., x_{k+1}\}$, parameter $\lambda > 0$
    Compute $U = [x_1 - x_0, ..., x_{k+1} - x_k]$
    Solve the linear system $(U^T U + \lambda I)z = \mathbf{1}$
    Set $c = z/(z^T \mathbf{1})$
**Output:** $\sum_{i=0}^{k} c_i x_i$, the approximation of the fixed point $x^*$

---

The computational complexity (with online updates or in batch mode) of the algorithm is $O(nk^2)$ and some strategies (batch and online) are discussed in the Appendix A.3. Note that the algorithm never calls the oracle $g(x)$. It means that, in an optimization context, the acceleration does not require $f(x)$ or $f'(x)$ to compute the extrapolation. Moreover, it does not need a priori information on the function, for example $L$ and $\mu$ (unlike Nesterov's method).

### 2.5 Convergence Bounds on Regularized AMPE

To fully characterize the convergence of our estimate sequence, we still need to bound the last term on the right-hand side of (13), namely $\| \sum_{i=0}^{k} c_i x_i - x^* \|_2$. A coarse bound can be provided using Chebyshev polynomials, however the norm of the Chebyshev's coefficients grows exponentially as $k$ grows. Here we refine this bound to better control the quality of our estimate.

Let $g(x^*) \preceq \sigma I$. Consider the following Chebyshev-like optimization problem, written

$$S(k, \alpha) \triangleq \min_{\{q \in \mathbb{R}_k[x]:\, q(1)=1\}} \left\{ \max_{x \in [0, \sigma]} \left( (1-x)q(x) \right)^2 + \alpha \|q\|_2^2 \right\}, \tag{16}$$

where $\mathbb{R}_k[x]$ is the ring of polynomials of degree at most $k$ and $q \in \mathbb{R}^{k+1}$ is the vector of coefficients of the polynomial $q(x)$. This problem can be solved exactly using a semi-definite solver because it can be reduced to a SDP program (see Appendix A.4 for the details of the reduction). Our main result below shows how $S(k, \alpha)$ bounds the error between our estimate of the optimum constructed using the iterates $\tilde{x}_i$ in (RMPE) and the optimum $x^*$ of problem (1).

**Proposition 2.6** *Let matrices $X = [x_0, x_1, ..., x_k]$, $\tilde{X} = [x_0, \tilde{x}_1, ..., \tilde{x}_k]$, $\mathcal{E} = (X - \tilde{X})$ and scalar $\kappa = \|(A - I)^{-1}\|_2$. Suppose $\tilde{c}_\lambda^*$ solves problem* (RMPE)

$$\begin{array}{ll} minimize & c^T (\tilde{U}^T \tilde{U} + \lambda I)c \\ subject\ to & \mathbf{1}^T c = 1 \end{array} \qquad \Rightarrow \qquad \tilde{c}_\lambda^* = \frac{(\tilde{U}^T \tilde{U} + \lambda I)^{-1} \mathbf{1}}{\mathbf{1}^T (\tilde{U}^T \tilde{U} + \lambda I)^{-1} \mathbf{1}} \tag{17}$$

*in the variable $c \in \mathbb{R}^{k+1}$, with parameters $\tilde{U} \in \mathbb{R}^{n \times (k+1)}$. Assume $A$ symmetric with $0 \preceq A \prec I$. Then*

$$\|\tilde{X} \tilde{c}_\lambda^* - x^*\|_2 \leq \left( \kappa^2 + \frac{1}{\lambda} \left( 1 + \frac{\|P\|_2}{\lambda} \right)^2 \left( \|\mathcal{E}\|_2 + \kappa \frac{\|P\|_2}{2\sqrt{\lambda}} \right)^2 \right)^{\frac{1}{2}} \left( S(k, \lambda/\|x_0 - x^*\|_2^2) \right)^{\frac{1}{2}} \|x_0 - x^*\|_2,$$

*with $P$ is defined in Corollary 2.5 and $S(k, \alpha)$ is defined in (16).*

We have that $S(k, \lambda/\|x_0 - x^*\|_2^2)^{\frac{1}{2}}$ is similar to the value $\mathcal{T}_k(\sigma)$ (see (8)) so our algorithm achieves a rate similar to the Chebyshev's acceleration up to some multiplicative scalar. We thus need to choose $\lambda$ so that this multiplicative scalar is not too high (while keeping $S(k, \lambda/\|x_0 - x^*\|_2^2)^{\frac{1}{2}}$ small).

We can analyze the behavior of the bound if we start close to the optimum. Assume

$$\|\mathcal{E}\|_2 = O(\|x_0 - x^*\|_2^2), \qquad \|U\|_2 = O(\|x_0 - x^*\|_2) \quad \Rightarrow \quad \|P\|_2 = O(\|x_0 - x^*\|_2^3).$$

This case is encountered when minimizing a smooth strongly convex function with Lipchitz-continuous Hessian using fixed-step gradient method (this case is discussed in details in the Appendix, section A.6). Also, let $\lambda = \beta\|P\|_2$ for $\beta > 0$ and $\|x_0 - x^*\|$ small. We can thus approximate the right parenthesis of the bound by

$$\lim_{\|x - x^*\|_2 \to 0} \left( \|\mathcal{E}\|_2 + \kappa \frac{\|P\|_2}{2\sqrt{\lambda}} \right) = \lim_{\|x - x^*\|_2 \to 0} \left( \|\mathcal{E}\|_2 + \kappa \frac{\sqrt{\|P\|_2}}{2\sqrt{\beta}} \right) = \frac{\kappa\sqrt{\|P\|_2}}{2\sqrt{\beta}}.$$

Therefore, the bound on the precision of the extrapolation is approximately equal to

$$\|\tilde{X}\tilde{c}_\lambda^* - x^*\|_2 \quad \lesssim \quad \kappa \left( 1 + \frac{(1 + \frac{1}{\beta})^2}{4\beta^2} \right)^{1/2} \sqrt{S\left( k, \frac{\beta\|P\|_2}{\|x_0 - x^*\|_2^2} \right)} \|x_0 - x^*\|_2$$

Also, if we use equation (8), it is easy to see that

$$\sqrt{S(k, 0)} \le \min_{\{q \in \mathbb{R}_k[x]: \, q(1) = 1\}} \max_{x \in [0, \sigma_1]} |q(x)| = \mathcal{T}_k(t(\sigma)) = \frac{2\zeta^k}{1 + \zeta^{2k}},$$

where $\zeta$ is defined in (9). So, when $\|x_0 - x^*\|_2$ is close to zero, the regularized version of AMPE tends to converge as fast as AMPE (see equation (10)) up to a small constant.

## 3   Numerical Experiments

We test our methods on a regularized logistic regression problem written

$$f(w) = \sum_{i=1}^m \log\left(1 + \exp(-y_i \xi_i^T w)\right) + \frac{\tau}{2}\|w\|_2^2,$$

where $\Xi = [\xi_1, ..., \xi_m]^T \in \mathbb{R}^{m \times n}$ is the design matrix and $y$ is a $\{-1, 1\}^n$ vector of labels. We used the *Madelon* UCI dataset, setting $\tau = 10^2$ (in order to have a ratio $L/\tau$ equal to $10^9$). We solve this problem using several algorithms, the fixed-step gradient method for strongly convex functions [6, Th. 2.1.15] using stepsize $2/(L + \mu)$, where $L = \|\Xi\|_2^2/4 + \tau$ and $\mu = \tau$, the accelerated gradient method for strongly convex functions [6, Th. 2.2.3] and our nonlinear acceleration of the gradient method iterates using RMPE in Proposition 2.6 with restarts.

This last algorithm is implemented as follows. We do $k$ steps (in the numerical experiments, $k$ is typically equal to 5) of the gradient method, then extrapolate a solution $\tilde{X}\tilde{c}_\lambda^*$ where $\tilde{c}_\lambda^*$ is computed by solving the RMPE problem (17) on the gradient iterates $\tilde{X}$, with regularization parameter $\lambda$ determined by a grid search. Then, this extrapolation becomes the new starting point of the gradient method. We consider it as one iteration of RMPE$k$ using $k$ gradient oracle calls. We also analyze the impact of an inexact line-search (described in Appendix A.7) performed after this procedure.

The results are reported in Figure 1. Using very few iterates, the solution computed using our estimate (a nonlinear average of the gradient iterates) are markedly better than those produced by the Nesterov-accelerated method. This is only partially reflected by the theoretical bound from Proposition 2.6 which shows significant speedup in some regions but remains highly conservative (see Figure 3 in section A.6). Also, Figure 2 shows us the impact of regularization. The AMPE process becomes unstable because of the condition number of matrix $M$, which impacts the precision of the estimate.

## 4   Conclusion and Perspectives

In this paper, we developed a method which is able to accelerate, under some regularity conditions, the convergence of a sequence $\{x_i\}$ without any knowledge of the algorithm which generates this sequence. The regularization parameter present in the acceleration method can be computed easily using some inexact line-search.

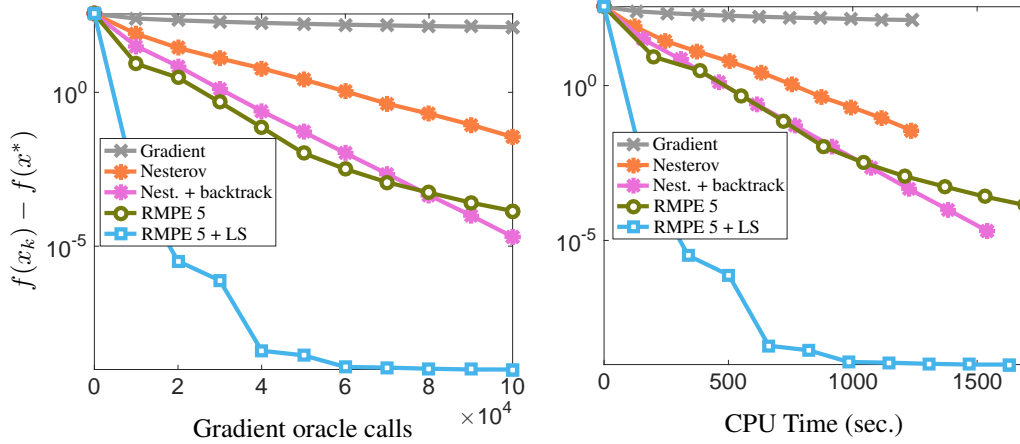

Figure 1: Solving logistic regression on *UCI Madelon dataset* (500 features, 2000 data points) using the gradient method, Nesterov's accelerated method and RMPE with $k = 5$ (with and without line search over the stepsize), with penalty parameter $\tau$ equal to $10^2$ (Condition number is equal to $1.2 \cdot 10^9$). Here, we see that our algorithm has a similar behavior to the conjugate gradient: unlike the Nesterov's method, where we need to provide parameters $\mu$ and $L$, the RMPE algorithm adapts himself in function of the spectrum of $g(x^*)$ (so it can exploit the good local strong convexity parameter), without any prior specification. We can, for example, observe this behavior when the global strong convexity parameter is bad but not the local one.

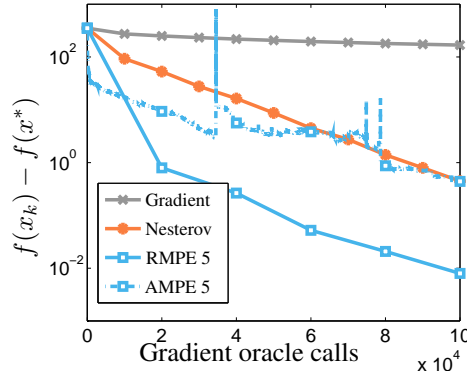

Figure 2: Logistic regression on *Madelon UCI Dataset*, solved using Gradient method, Nesterov's method and AMPE (i.e. RMPE with $\lambda = 0$). The condition number is equal to $1.2 \cdot 10^9$. We see that without regularization, AMPE is unstable because $\|(\tilde{U}^T\tilde{U})^{-1}\|_2$ is huge (see Proposition 2.3).

The algorithm itself is simple. By solving only a small linear system we are able to compute a good estimate of the limits of the sequence $\{x_i\}$. Also, we showed (using the gradient method on logistic regression) that the strategy which consists in alternating the algorithm and the extrapolation method can lead to impressive results, improving significantly the rate of convergence.

Future work will consist in improving the performance of the algorithm by exploiting the structure of the noise matrix $E$ in some cases (for example, using the gradient method, the norm of the column $E_k$ in the matrix $E$ is decreasing when $k$ grows), extending the algorithm to the constrained case, the stochastic case and to the non-symmetric. We will also try to refine the term (16) present in the theoretical bound.

**Acknowledgment.** The research leading to these results has received funding from the European Union's Seventh Framework Programme (FP7-PEOPLE-2013-ITN) under grant agreement nº 607290 SpaRTaN, as well as support from ERC SIPA and the chaire *Économie des nouvelles données* with the *data science* joint research initiative with the *fonds AXA pour la recherche*.

## Footnotes

[1] A monic polynomial is a univariate polynomial in which the coefficient of highest degree is equal to 1.

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
