[Supplementary Material]

# A    Supplementary material

This section contains proofs and additional details on some of the material discussed above.

## A.1    Proofs of minimal polynomial normalization

**Lemma A.1** *Let $p(x)$ be the minimal polynomial of the matrix $M$. Assume that $M$ does not have the eigenvalue $1$. Then we can always normalize $p$ by the sum of its coefficients, $p(1)$.*

**Proof.** Let $p$ be the minimal polynomial of $M$. Then for all eigenvalue $\lambda_i$ of $M$, we have $p(\lambda_i) = 0$. Since $p$ is minimal, we cannot find $q$ with $\deg q < \deg p$ satisfying the above condition.

Now, assume by contradiction that $p(1) = 0$. It means that $1$ is a root of $p$, so $p$ can be written

$$p(x) = (1 - x)q(x).$$

Because $1$ is not an eigenvalue of $M$, we have $p(\lambda_i) = 0$ if and only if $q(\lambda_i) = 0$. However $\deg q < \deg p$, so $p$ is not minimal. ■

## A.2    Proofs of regularization results

**Proposition A.2** *Let $c^*$ be the optimal solution to* (AMPE)

$$c^* = \underset{\mathbf{1}^T c = 1}{\operatorname{argmin}} \; \|Uc\|_2$$

*for some matrix $U \in \mathbb{R}^{n,k}$. Suppose $U$ becomes $\tilde{U} = U + E$ and write $c^* + \Delta c$ the perturbed solution to* (AMPE)*, then*

$$\Delta c = -\left(I - \frac{M^{-1}\mathbf{1}\mathbf{1}^T}{\mathbf{1}^T M^{-1}\mathbf{1}}\right) M^{-1}(U^T E + E^T U + E^T E)c^* \tag{18}$$

*where $M = (U + E)^T (U + E)$ and*

$$\left(I - \frac{M^{-1}\mathbf{1}\mathbf{1}^T}{\mathbf{1}^T M^{-1}\mathbf{1}}\right)$$

*is a projector of rank $k - 1$.*

**Proof.** Let $\mu$ be the dual variable corresponding to the equality constraint. Both $c^* + \Delta c$ and $\mu^* + \Delta\mu$ must satisfy the KKT system

$$\begin{bmatrix} 2M & \mathbf{1} \\ \mathbf{1}^T & 0 \end{bmatrix} \begin{pmatrix} c^* + \Delta c \\ \mu^* + \Delta\mu \end{pmatrix} = \begin{pmatrix} 0 \\ 1 \end{pmatrix},$$

writing $P = U^T E + E^T U + E^T E$, this means again

$$\begin{aligned} \begin{bmatrix} 2M & \mathbf{1} \\ \mathbf{1}^T & 0 \end{bmatrix} \begin{pmatrix} c^* + \Delta c \\ \mu^* + \Delta\mu \end{pmatrix} &= \begin{bmatrix} 2P & 0 \\ 0 & 0 \end{bmatrix} \begin{pmatrix} c^* + \Delta c \\ \mu^* + \Delta\mu \end{pmatrix} + \begin{bmatrix} 2U^T U & \mathbf{1} \\ \mathbf{1}^T & 0 \end{bmatrix} \begin{pmatrix} c^* + \Delta c \\ \mu^* + \Delta\mu \end{pmatrix} \\ &= \begin{pmatrix} 2P(c^* + \Delta c) \\ 0 \end{pmatrix} + \begin{bmatrix} 2U^T U & \mathbf{1} \\ \mathbf{1}^T & 0 \end{bmatrix} \begin{pmatrix} \Delta c \\ \Delta\mu \end{pmatrix} + \begin{pmatrix} 0 \\ 1 \end{pmatrix} \end{aligned}$$

hence

$$\begin{bmatrix} 2M & \mathbf{1} \\ \mathbf{1}^T & 0 \end{bmatrix} \begin{pmatrix} \Delta c \\ \Delta\mu \end{pmatrix} = \begin{pmatrix} -2Pc^* \\ 0 \end{pmatrix}$$

The block matrix can be inverted explicitly, with

$$\begin{bmatrix} 2M & \mathbf{1} \\ \mathbf{1}^T & 0 \end{bmatrix}^{-1} = \frac{1}{\mathbf{1}^T M^{-1}\mathbf{1}} \begin{bmatrix} \frac{1}{2}M^{-1}\left((\mathbf{1}^T M^{-1}\mathbf{1})I - \mathbf{1}\mathbf{1}^T M^{-1}\right) & M^{-1}\mathbf{1} \\ \mathbf{1}^T M^{-1} & -2 \end{bmatrix}$$

leading to an expression of $\Delta c$ and $\Delta \mu$ in terms of $c^*$ and $\mu^*$:

$$\begin{pmatrix} \Delta c \\ \Delta \mu \end{pmatrix} = \frac{1}{\mathbf{1}^T M^{-1} \mathbf{1}} \begin{bmatrix} \frac{1}{2} M^{-1} \left( (\mathbf{1}^T M^{-1} \mathbf{1}) I - \mathbf{1}\mathbf{1}^T M^{-1} \right) & M^{-1}\mathbf{1} \\ \mathbf{1}^T M^{-1} & -2 \end{bmatrix} \begin{pmatrix} -2Pc^* \\ 0 \end{pmatrix}$$

After some simplification, we get

$$\Delta c = -\left( I - \frac{M^{-1}\mathbf{1}\mathbf{1}^T}{\mathbf{1}^T M^{-1}\mathbf{1}} \right) M^{-1} Pc^* = -W M^{-1} Pc^*$$

where $W$ is a projector of rank $k-1$, which is the desired result. ∎

**Lemma A.3** *Let $c^*_\lambda$ be the optimal solution of problem (RMPE). Then*

$$c^*_\lambda = \frac{(U^T U + \lambda I)^{-1}\mathbf{1}}{\mathbf{1}^T (U^T U + \lambda I)^{-1}\mathbf{1}} \tag{19}$$

*Therefore,*

$$\|c^*_\lambda\|_2 \le \sqrt{\frac{\lambda + \|U\|_2^2}{k\lambda}} \tag{20}$$

**Proof.** Let $c^*_\lambda$ the optimal solution of the primal and $\nu^*_\lambda$ the optimal dual variable of problem (RMPE). Let $M_\lambda = U^T U + \lambda I$. Then both $c^*_\lambda$ and $\nu^*_\lambda$ must satisfy the KKT system:

$$\begin{bmatrix} 2M_\lambda & \mathbf{1} \\ \mathbf{1}^T & 0 \end{bmatrix} \begin{pmatrix} c^*_\lambda \\ \mu^*_\lambda \end{pmatrix} = \begin{pmatrix} 0 \\ 1 \end{pmatrix}.$$

We have thus

$$\begin{pmatrix} c^*_\lambda \\ \mu^*_\lambda \end{pmatrix} = \begin{bmatrix} 2M_\lambda & \mathbf{1} \\ \mathbf{1}^T & 0 \end{bmatrix}^{-1} \begin{pmatrix} 0 \\ 1 \end{pmatrix}.$$

The block matrix can be inverted explicitly:

$$\begin{bmatrix} 2M_\lambda & \mathbf{1} \\ \mathbf{1}^T & 0 \end{bmatrix}^{-1} = \frac{1}{\mathbf{1}^T M_\lambda^{-1}\mathbf{1}} \begin{bmatrix} \frac{1}{2} M_\lambda^{-1}((\mathbf{1}^T M_\lambda^{-1}\mathbf{1})I - \mathbf{1}\mathbf{1}^T M_\lambda^{-1}) & M_\lambda^{-1}\mathbf{1} \\ \mathbf{1}^T M_\lambda^{-1} & -2 \end{bmatrix},$$

leading to

$$c^*_\lambda = \frac{M_\lambda^{-1}\mathbf{1}}{\mathbf{1}^T M_\lambda^{-1}\mathbf{1}}.$$

Since

$$\|M_\lambda^{-1}\|_2 \le \frac{1}{\sigma_{\min}(U^T U) + \lambda} \le \frac{1}{\lambda}$$

and

$$\mathbf{1}^T M_\lambda^{-1}\mathbf{1} \ge \frac{\|\mathbf{1}\|^2}{\sigma_{\max}(M_\lambda)} \ge \frac{k}{\|U\|_2^2 + \lambda}$$

We obtain

$$\|c^*_\lambda\|_2 = \frac{\|M_\lambda^{-1/2} M_\lambda^{-1/2}\mathbf{1}\|_2}{\mathbf{1}^T M_\lambda^{-1}\mathbf{1}} \le \frac{\|M_\lambda^{-1}\|_2^{1/2}}{\sqrt{\mathbf{1}^T M_\lambda^{-1}\mathbf{1}}} \le \sqrt{\frac{\lambda + \|U\|_2^2}{k\lambda}}$$

which is the desired result. ∎

### A.3   Computational Complexity for RMPE Algorithm

In Algorithm 1, computing the coefficients $\tilde{c}^*_\lambda$ means solving the $k \times k$ system $(\tilde{U}^T \tilde{U} + \lambda I)z = \mathbf{1}$. We then get $\tilde{c}^*_\lambda = z/(\mathbf{1}^T z)$. This can be done in both batch and online mode.

**Online updates.** Here, we receive the vectors $u_i$ one by one from the optimization algorithm. In this case, we perform low-rank updates on the Cholesky factorization of the system matrix. At iteration $i$, we have the Cholesky factorization $LL^T = \tilde{U}^T\tilde{U} + \lambda I$. We receive a new vector $u_+$ and we want

$$L_+L_+^T = \begin{bmatrix} L & 0 \\ a^T & b \end{bmatrix} \begin{bmatrix} L^T & a \\ 0 & b \end{bmatrix} = \begin{bmatrix} \tilde{U}^T\tilde{U} + \lambda I & \tilde{U}^T u_+ \\ (\tilde{U}^T u_+)^T & u_+^T u_+ + \lambda \end{bmatrix} \Rightarrow a = L^{-1}\tilde{U}^T u_+, \ b = a^T a + \lambda.$$

The complexity of this update is thus $O(in + i^2)$, i.e. the matrix-vector multiplication of $\tilde{U}^T u_+$ and solving the triangular system. Since we need to do it $k$ times, the final complexity is thus $O(nk^2 + k^3)$. Notice that, at the end, it takes only $O(k^2)$ iteration to solve the system $LL^T z = \mathbf{1}$.

**Batch mode.** The complexity is divided in two parts: First, we need to build the linear system itself. Since $U \in \mathbb{R}^{n\times k}$, it takes $O(nk^2)$ flops to perform the multiplication. Then we need to solve the linear system $(\tilde{U}^T\tilde{U} + \lambda I)z = \mathbf{1}$ which can be done by a direct solver like Gaussian elimination (if $k$ is small) or Cholesky factorization, or using an iterative method like conjugate gradient method. It takes $O(k^3)$ flops to solve the linear system in the worst case, meaning that the complexity at the end is $O(nk^2 + k^3)$. In practice, the eigenvalues of the system tend to be clustered around $\lambda$, which means that the conjugate gradient solver converges very quickly to a good solution.

## A.4  Regularized Chebychev Polynomials

We first briefly recall basic results on Sum of Squares (SOS) polynomials and moment problems [25, 26, 27], which will allow us to formulate problem (16) as a (tractable) semidefinite program. A univariate polynomial is positive if and only if it is a sum of squares. Furthermore, if we let $m(x) = (1, x, \ldots, x^k)^T$ we have, for any $p(x) \in \mathbb{R}_{2k}[x]$,

$$p(x) \geq 0, \text{ for all } x \in \mathbb{R}$$
$$\Updownarrow$$
$$p(x) = m(x)^T C m(x), \text{ for some } C \succeq 0,$$

which means that checking if a polynomial is positive on the real line is equivalent to solving a linear matrix inequality (see e.g. [28, §4.2] for details). We can thus write the problem of computing the maximum of a polynomial over the real line as

$$\begin{aligned} \text{minimize} \quad & t \\ \text{subject to} \quad & t - p(x) = m(x)^T C m(x), \quad \text{for all } x \in \mathbb{R} \\ & C \succeq 0, \end{aligned} \tag{21}$$

which is a semidefinite program in the variables $p \in \mathbb{R}^{k+1}$, $C \in \mathbf{S}_{k+1}$ and $t \in \mathbb{R}$, because the first contraint is equivalent to a set of linear equality constraints. Then, showing that $p(x) \geq 0$ on the segment $[0, \sigma]$ is equivalent to showing that the rational fraction $p\left(\sigma x^2/(1+x^2)\right)$ is positive on the real line, or equivalently, that the polynomial is positive on the real line. Overall, this implies that problem (16) can be written

$$\begin{aligned} S(k, \alpha) = \quad \text{min.} \quad & t^2 + \alpha^2 \|q\|_2^2 \\ \text{s.t.} \quad & (1+x^2)^{k+1}\left(\left(1 - \frac{\sigma x^2}{1+x^2}\right) q\left(\frac{\sigma x^2}{1+x^2}\right)\right) = m(x)^T C m(x), \quad \text{for all } x \in \mathbb{R} \\ & \mathbf{1}^T q = 1, \ C \succeq 0, \end{aligned}$$
$$\tag{22}$$

which is a semidefinite program in the variables $q \in \mathbb{R}^{k+1}$, $C \in \mathbf{S}_{k+2}$ and $t \in \mathbb{R}$.

## A.5  Proof of the convergence result

**Proposition A.4** *Let matrices* $X = [x_0, x_1, \ldots, x_k]$, $\tilde{X} = [x_0, \tilde{x}_1, \ldots, \tilde{x}_k]$, $\mathcal{E} = (X - \tilde{X})$ *and scalar* $\kappa = \|(A - I)^{-1}\|_2$. *Suppose* $\tilde{c}_\lambda^*$ *solves problem* (RMPE)

$$\begin{aligned} \text{minimize} \quad & c^T(\tilde{U}^T\tilde{U} + \lambda I)c \\ \text{subject to} \quad & \mathbf{1}^T c = 1 \end{aligned} \quad \Rightarrow \quad c_\lambda^* = \frac{(\tilde{U}^T\tilde{U} + \lambda I)^{-1}\mathbf{1}}{\mathbf{1}^T(\tilde{U}^T\tilde{U} + \lambda I)^{-1}\mathbf{1}} \tag{23}$$

in the variable $c \in \mathbb{R}^{k+1}$, with parameters $\tilde{U} \in \mathbb{R}^{n \times (k+1)}$. Assume $A$ symmetric with $0 \preceq A \prec I$. Then

$$\|\tilde{X}\tilde{c}_\lambda^* - x^*\|_2 \leq \left( \kappa^2 + \frac{1}{\lambda} \left( 1 + \frac{\|P\|_2}{\lambda} \right)^2 \left( \|\mathcal{E}\|_2 + \kappa \frac{\|P\|_2}{2\sqrt{\lambda}} \right)^2 \right)^{\frac{1}{2}} \left( S(k, \lambda/\|x_0 - x^*\|_2^2) \right)^{\frac{1}{2}} \|x_0 - x^*\|_2$$

with $P$ is defined in Corollary 2.5 and $S(k, \alpha)$ is defined in (16).

**Proof.** Let us write the error decomposition (13) in matrix format:

$$\|\tilde{X}\tilde{c}_\lambda^* - x^*\|_2 \leq \|Xc_\lambda^* - x^*\|_2 + \|(X - X^*)\Delta c\|_2 + \|\mathcal{E}\tilde{c}_\lambda^*\|_2.$$

The first term can be bounded as follow:

$$\begin{aligned}
\|Xc_\lambda^* - x^*\|_2 &\leq& \kappa\|Uc_\lambda^*\|_2 \\
&\leq& \kappa\sqrt{\|Uc_\lambda^*\|_2^2 + (\lambda - \lambda)\|c_\lambda^*\|_2^2} \\
&\leq& \kappa\sqrt{\|(A - I)p(A)\|_2^2\|x_0 - x^*\|_2^2 + \lambda\|c_\lambda^*\|_2^2 - \lambda\|c_\lambda^*\|_2^2} \\
&\leq& \kappa\sqrt{S(k, \lambda/\|x_0 - x^*\|_2^2)\|x_0 - x^*\|_2^2 - \lambda\|c_\lambda^*\|_2^2}.
\end{aligned}$$

The second one becomes, if we use Corollary 2.5,

$$\begin{aligned}
\|(X - X^*)\Delta c_\lambda^*\|_2 &\leq& \kappa\|U\Delta c_\lambda^*\|_2 \\
&\leq& \kappa\|U(U^T U + \lambda I + P)^{-1}\tilde{W}^T P\|_2\|c_\lambda^*\|_2 \\
&\leq& \kappa\|U(U^T U + \lambda I + P)^{-1}\|_2\|P\|_2\|c_\lambda^*\|_2.
\end{aligned}$$

Let us write $(U^T U + \lambda I + P)^{-1} = [(U^T U + \lambda I)^{-1} + S]$ for some perturbation $S$. Indeed,

$$((U^T U + \lambda I)^{-1} + S)(U^T U + \lambda I + P) = I,$$

which leads to

$$S = -(U^T U + \lambda I)^{-1}P(U^T U + \lambda I + P)^{-1}.$$

If we plug this expression in $\|U(U^T U + \lambda I + P)^{-1}\|_2$ we obtain

$$\begin{aligned}
\|U(U^T U + \lambda I + P)^{-1}\|_2 &=& \|U(U^T U + \lambda I)^{-1}(I - P(U^T U + \lambda I + P)^{-1})\|_2 \\
&\leq& \|U(U^T U + \lambda I)^{-1}\|_2 \left( 1 + \|P\|_2\|(U^T U + \lambda I + P)^{-1}\|_2 \right) \\
&\leq& \frac{\sigma}{\sigma^2 + \lambda} \left( 1 + \frac{\|P\|_2}{\lambda} \right).
\end{aligned}$$

For any value of $\sigma \in [\sigma_{\min}^{1/2}(U^T U), \sigma_{\max}^{1/2}(U^T U)]$. The maximum is attained at $\sigma = \sqrt{\lambda}$, so it becomes

$$\|U(U^T U + \lambda I + P)^{-1}\|_2 \leq \frac{1}{2\sqrt{\lambda}} \left( 1 + \frac{\|P\|_2}{\lambda} \right).$$

So the second term can be bounded by

$$\|(X - X^*)\Delta c_\lambda^*\|_2 \leq \kappa\frac{\|P\|_2}{2\sqrt{\lambda}} \left( 1 + \frac{\|P\|_2}{\lambda} \right) \|c_\lambda^*\|.$$

The third term can be bounded as follow:

$$\begin{aligned}
\|\mathcal{E}\tilde{c}_\lambda^*\|_2 &\leq& \|\mathcal{E}\|_2(\|c_\lambda^*\|_2 + \|\Delta c_\lambda^*\|_2) \\
&\leq& \|\mathcal{E}\|_2 \left( 1 + \frac{\|P\|_2}{\lambda} \right) \|c_\lambda^*\|_2.
\end{aligned}$$

If we combine all bounds, we obtain

$$\|\tilde{X}\tilde{c}_\lambda^* - x^*\|_2^2 \leq \kappa\sqrt{S(k, \lambda/\|x_0 - x^*\|_2^2)\|x_0 - x^*\|_2^2 - \lambda\|c_\lambda^*\|_2^2} + \|c_\lambda^*\|_2 \left( 1 + \frac{\|P\|_2}{\lambda} \right) \left( \|\mathcal{E}\|_2 + \kappa\frac{\|P\|_2}{2\sqrt{\lambda}} \right).$$

We will now find the value of $\|c_\lambda^*\|_2$ which maximize the bound. For more simplicity, let us write the bound using parameters $a$, $b$ and $c = \|c_\lambda^*\|_2$:

$$\kappa\sqrt{a^2 - \lambda c^2} + bc.$$

We want to solve

$$\max_{c:0 \leq c \leq (a/\sqrt{\lambda})} \kappa\sqrt{a^2 - \lambda c^2} + bc,$$

and the solution is given by

$$c = \frac{a}{\sqrt{\lambda}} \frac{b}{\sqrt{\kappa^2\lambda + b^2}} \in \left[0, \frac{a}{\sqrt{\lambda}}\right].$$

The optimal value becomes

$$\max_{c:0 \leq c \leq (a/\sqrt{\lambda})} \kappa\sqrt{a^2 - \lambda c^2} + bc = \frac{a}{\sqrt{\lambda}}\sqrt{\kappa^2\lambda + b^2}.$$

In other words, if we replace $a$, $b$ and $c$, we have

$$\|\tilde{X}\tilde{c}_\lambda^* - x^*\|_2 \leq \sqrt{S(k, \lambda/\|x_0 - x^*\|_2^2)}\|x_0 - x^*\|_2\sqrt{\kappa^2 + \frac{1}{\lambda}\left(1 + \frac{\|P\|_2}{\lambda}\right)^2\left(\|\mathcal{E}\|_2 + \kappa\frac{\|P\|_2}{2\sqrt{\lambda}}\right)^2},$$

which is the desired result. ∎

### A.6 Explicit bounds for the gradient method

Here, we make our bounds explicit in the case where we use the simple gradient method for smooth and strongly convex function with Lipchitz-continuous Hessian. In this scenario, the fixed point function becomes

$$\tilde{x}_{k+1} = \tilde{x}_k - \frac{1}{L}f'(\tilde{x}_k)$$

where $\mu I \preceq f''(x) \preceq LI$ (and we assume the bounds thigh at $x = x^*$ for simplicity). Moreover,

$$\|f''(y) - f''(x)\|_2 \leq M\|y - x\|_2$$

We have thus $A = I - \frac{1}{L}f''(x^*)$, meaning that $\|A\|_2 \leq 1 - \frac{\mu}{L}$. The rate of this method is

$$\|\tilde{x}_k - x^*\|_2 \leq \left(\sqrt{\frac{L-\mu}{L+\mu}}\right)^k\|x_0 - x^*\|_2 = r^k\|x_0 - x^*\|_2$$

Notice that this is not the optimal fixed-step gradient method, however it allows us a much simpler analysis. Now let us bound $\|\tilde{X} - X^*\|_2$, $\|U\|_2$ and $\|E\|_2$. Indeed,

$$
\begin{aligned}
\|\tilde{X} - X^*\|_2 &\leq \sum_{i=0}^{k}\|\tilde{x}_i - x^*\|_2 \\
&= \frac{1 - r^k}{1 - r}\|x_0 - x^*\|_2 \\
\|U\|_2 &\leq \|A - I\|_2\sum_{i=0}^{k}\|x_i - x^*\|_2 \\
&\leq \sum_{i=0}^{k}\|A\|^i\|x_0 - x^*\|_2 \\
&\leq \frac{1 - \|A\|_2^k}{1 - \|A\|_2}\|x_0 - x^*\|_2 \\
&\leq \frac{L}{\mu}\left(1 - \left(1 - \frac{\mu}{L}\right)^k\right)\|x_0 - x^*\|_2
\end{aligned}
$$

$$\tilde{x}_{i+1} - x_{i+1} = \tilde{x}_i - \frac{1}{L}f'(\tilde{x}_i) - x_i + \frac{1}{L}f''(x^*)(x_i - x^*)$$

$$= \tilde{x}_i - x_i - \frac{1}{L}(f'(\tilde{x}_i) - f''(x^*)(x_i - x^*))$$

$$= (I - \frac{f''(x^*)}{L})(\tilde{x}_i - x_i) - \frac{1}{L}(f'(\tilde{x}_i) - f''(x^*)(\tilde{x}_i - x^*))$$

Since our function has a Lipchitz-continuous Hessian, it is possible to show that ([6], Lemma 1.2.4)

$$\|f'(\tilde{y}) - f'(x) - f''(x)(\tilde{y} - x)\|_2 \le \frac{M}{2}\|y - x\|^2$$

We can thus bound the norm of the error at the $i^{\text{th}}$ iteration:

$$\|x_{i+1} - \tilde{x}_{i+1}\|_2 \le \left\|I - \frac{f''(x^*)}{L})\right\|_2 \|x_i - \tilde{x}_i\|_2 + \frac{1}{L}\|f'(\tilde{x}_i) - f''(x^*)(\tilde{x}_i - x^*)\|_2$$

$$= \left\|I - \frac{f''(x^*)}{L})\right\|_2 \|x_i - \tilde{x}_i\|_2 + \frac{1}{L}\|f'(\tilde{x}_i) - f'(x^*) - f''(x^*)(\tilde{x}_i - x^*)\|_2$$

$$\le (1 - \frac{\mu}{L})\|x_i - \tilde{x}_i\|_2 + \frac{M}{2L}\|\tilde{x}_i - x^*\|_2^2$$

$$\le (1 - \frac{\mu}{L})\|x_i - \tilde{x}_i\|_2 + \frac{M}{2L}(r^2)^i\|x_0 - x^*\|_2^2$$

$$= \frac{M}{2L}\sum_{j=1}^{i}(1 - \frac{\mu}{L})^{i-j}(r^2)^j\|x_0 - x^*\|_2^2$$

The sum starts at $j = 1$ because, by definition, $\|e_0\|_2 = 0$. In order to have a simpler analysis, let us use the fact that

$$\frac{r^2}{1 - \frac{\mu}{L}} = \frac{1}{1 + \frac{\mu}{L}} < 1$$

We can bound $\|x_{i+1} - \tilde{x}_{i+1}\|_2$ with a simpler expression:

$$\|x_{i+1} - \tilde{x}_{i+1}\|_2 \le (1 - \frac{\mu}{L})^i \frac{M}{2L}\sum_{j=1}^{i}\left(\frac{r^2}{1 - \frac{\mu}{L}}\right)^j\|x_0 - x^*\|_2^2$$

$$\le (1 - \frac{\mu}{L})^i \frac{M}{2L}\left(\sum_{j=0}^{i}\left(\frac{r^2}{1 - \frac{\mu}{L}}\right)^j\right)\|x_0 - x^*\|_2^2$$

$$= (1 - \frac{\mu}{L})^i \frac{M}{2L}\left(\sum_{j=0}^{i}\left(\frac{1}{1 + \frac{\mu}{L}}\right)^j\right)\|x_0 - x^*\|_2^2$$

$$= (1 - \frac{\mu}{L})^i \frac{M}{2L}\left(\frac{1 - \left(\frac{1}{1+\frac{\mu}{L}}\right)^i}{1 - \frac{1}{1+\frac{\mu}{L}}}\right)\|x_0 - x^*\|_2^2$$

$$= \frac{M}{2L}\left(\frac{(1 - \frac{\mu}{L})^i - \left(\frac{1-\frac{\mu}{L}}{1+\frac{\mu}{L}}\right)^i}{1 - \frac{1}{1+\frac{\mu}{L}}}\right)\|x_0 - x^*\|_2^2$$

$$= \left(1 + \frac{L}{\mu}\right)\frac{M}{2L}\left((1 - \frac{\mu}{L})^i - \left(\frac{1-\frac{\mu}{L}}{1+\frac{\mu}{L}}\right)^i\right)\|x_0 - x^*\|_2^2$$

Figure 3: *Left:* Relative value for the regularization parameter $\lambda$ used in the theoretical bound. *Right:* Convergence speedup relative to gradient, for Nesterov's accelerated method and the theoretical RMPE bound in Proposition 2.6. We see that our algorithm performs a significant speedup (even in comparison with Nesterov's method) when $k$ is well chosen.

If we sum all the errors we get

$$
\begin{aligned}
\|\mathcal{E}\|_2 \;&\leq\; \sum_{i=0}^{k} \|x_i - \tilde{x}_i\|_2 \\
&\leq\; \left(1 + \frac{L}{\mu}\right) \frac{M}{2L} \left( \sum_{i=0}^{k} (1 - \tfrac{\mu}{L})^i - \sum_{i=0}^{k} \left(\frac{1 - \frac{\mu}{L}}{1 + \frac{\mu}{L}}\right)^i \right) \|x_0 - x^*\|_2^2 \\
&=\; \left(1 + \frac{L}{\mu}\right) \frac{M}{2L} \left( \frac{L}{\mu}\left(1 - \left(1 - \tfrac{\mu}{L}\right)^k\right) - \frac{L}{2\mu}\left(1 - \left(\frac{1 - \frac{\mu}{L}}{1 + \frac{\mu}{L}}\right)^k\right) \right) \|x_0 - x^*\|_2^2 \\
&\leq\; \left(1 + \frac{L}{\mu}\right)^2 \frac{M}{2L} \left( 1 - \left(1 - \tfrac{\mu}{L}\right)^k - \frac{1}{2}\left(1 - \left(\frac{1 - \frac{\mu}{L}}{1 + \frac{\mu}{L}}\right)^k\right) \right) \|x_0 - x^*\|_2^2 \\
&=\; \left(1 + \frac{L}{\mu}\right)^2 \frac{M}{2L} \left( \frac{1}{2} - \left(1 - \tfrac{\mu}{L}\right)^k + \frac{1}{2}\left(\frac{1 - \frac{\mu}{L}}{1 + \frac{\mu}{L}}\right)^k \right) \|x_0 - x^*\|_2^2
\end{aligned}
$$

We can also deduce that

$$\|\tilde{U} - U\|_2 = \|E\|_2 \leq 2\|\mathcal{E}\|_2 = \|\tilde{X} - X\|_2$$

Let us fix $\mu$, $L$, $M$ and $\|x_0 - x^*\|_2$ to some values:

- $L = 100$,
- $\mu = 10$,
- $M = 10^{-1}$,
- $\|x_0 - x^*\|_2 = 10^{-4}$.

We also decided to put $\lambda = \|P\|_2$.

In figure 3 (left) we can see the relative value for $\lambda$ (i.e. $\|P\|_2/\|x_0 - x^*\|^2$) using the above parameters. In this case, we can compute the rate of convergence of the AMPE method using proposition 2.6. This rate of convergence is showed in figure 3 (right).

## A.7  Additional numerical experiments

We test our methods on a regularized logistic regression problem written

$$f(w) = \sum_{i=1}^{m} \log\left(1 + \exp(-y_i \xi_i^T w)\right) + \frac{\tau}{2}\|w\|_2^2,$$

Optimization is done on dataset Madelon (500 features, 4400 points), sido0 (4932 features, 12678 points) and sonar (60 features, 208 points) with different values of $\tau$. The starting point is always $w = 0$. The optimization is done on raw data. We compare different algorithms:

- fixed-step gradient method with stepsize = $2/(L + \mu)$,
- Nesterov's method for strongly convex functions with fixed coefficients,
- Nesterov's method using backtracking strategy on the smoothness parameter $L$,
- RMPE with $k = 5$ (i.e. RMPE5), where $\lambda$ is found using grid search,
- RMPE5 using grid search for $\lambda$ and line-search for the stepsize.

The line-search on the stepsize works as follow. We compute an extrapolation $\bar{x} =$ RMPE$(X, k, \lambda)$ using iterates $X = [x_0, ..., x_k]$, then we find a coefficient $c \in \{1, 2, 4, ..., 2^i, ...\}$ which minimize the objective function $f(x_0 + c(\bar{x} - x_0))$. If we assume that the computation of the objective function value is much cheaper than the computation of its gradient, then this line-search has almost no impact on the total complexity of the algorithm.

In all experiments, we compare the rate of convergence of these methods in function of the number of gradient oracle call ot in function of time.

Figure 4: Logistic regression on Madelon dataset, with $\tau = 10^7$ (condition number $= 6 \cdot 10^3$).

Figure 5: Logistic regression on Madelon dataset, with $\tau = 10^{-3}$ (condition number $= 6 \cdot 10^{13}$).

Figure 6: Logistic regression on sonar dataset, with $\tau = 10^{-1}$ (condition number $= 7 \cdot 10^3$)

Figure 7: Logistic regression on sonar dataset, with $\tau = 10^{-6}$ (condition number $= 7 \cdot 10^8$)

Figure 8: Logistic regression on sido0 dataset, with $\tau = 10^2$ (condition number $= 1.5 \cdot 10^5$)