[Reviews · NeurIPS 2016]

Reviewer 1

Summary

The paper provides a generic strategy to accelerate iterative algorithms by estimating the optimal solution as a weighted combination of the iterates seen so far. The coefficients appearing in this combination are estimated by regularizing the natural linear estimator. When applied to strongly-convex, smooth functions this method recovers the asymptotic rate of the optimal method.

Qualitative Assessment

TECHNICAL QUALITY: The proofs are sound and the high-level presentation is clear. I consider the technical contribution both solid and interesting, as it connects iterative methods and numerical analysis in a novel way. However, I am unconvinced by the experimental evaluation. In particular, the RMPE algorithm proposed by the authors is compared with fixed-step-size versions of gradient descent and accelerated gradient descent. It seems to me that RMPE is inherently adaptive, as it attempts to estimate the best possible set of weights given the current iterates. I believe a more practical and fair comparison would involve adaptive versions of gradient descent and Nesterov averaging. A number of such heuristics exists and some theoretical results have been showed for unconstrained problems. In my experience evaluating the performance of Nesterov acceleration, the use of even very simple heuristics to adaptively reweigh the dual lower bound constructed by Nesterov leads to often dramatic improvements in performance. Hence, I am unsurprised that an adaptive method like RMPE would enjoy the same advantages. NOVELTY: I find the connection between the numerical techniques to extrapolate limits of sequences in this paper and the study of accelerated methods very interesting, as improving our understanding of accelerated methods is still an active area of research. However, I feel that this paper is still immature in appreciating and leveraging the depth of this connection. In particular, there have been a number of works on interpreting Nesterov's algorithm (e.g. Allen-Zhu and Orecchia; Bubeck, Lee and SIngh; Wibisono, WIlson and Jordan) that seem possibly related. In particular, I think that Allen-Zhu and Orecchia interpret Nesterov's acceleration as a combination of a primal and dual step, where the dual method maintains weights on the gradient iterates seen by the primal method. For the general proof, these weights are supposed to be uniform, but adaptive choices can lead to better performance in practice. IMPACT: This will have theoretical impact, but it could have much more if the ideas were placed in better context with respect to our current understanding of acceleration. CLARITY: While the authors clearly attempt to provide guidance to the reader in the form of high level comments and explanations, the paper covers a lot of material and is at times extremely dense. This is not helped by a) the presence of a number of typos (check spelling, e.g. line 228), b) the failed compilation of section references from the Latex source, c) the appearance of undefined or unquantified variable (line 150: what is c^*? It is only defined later. equation (16): what is sigma_1 in this context?). In the context of the empirical analysis, an algorithm NesterovConv appears in all the figures as the best performer, but it is not described anywhere. Based on reading the text, shouldn't this be some variants of RMPE? I may have missed something, so I would be happy to be corrected on this.

Confidence in this Review

3-Expert (read the paper in detail, know the area, quite certain of my opinion)


Reviewer 2

Summary

Extrapolation, i.e., constructing a new, potentially better estimate from existing ones, is a ubiquitous tool in numerical analysis. This work applied this old idea to optimization, to construct on the fly a better iterate. A regularized version (by adding ell_2 regularization) of the classic minimal polynomial extrapolation is proposed. The authors provided some crude asymptotic error analysis. Limited experiments against Nesterov's accelerated gradient and the conjugate gradient are provided, and the results seems to favor the proposed method.

Qualitative Assessment

===== added after author response ===== The authors' rebuttal was somewhat helpful. There are a few points that I'd like to emphasize again so that the authors would prepare a better version. "asymptotic analysis:" In line 199 and line 209, in order to deduce a meaningful bound from Proposition 2.6, you have to take the limit x_0 - x* --> 0 (and possibly k large), right? To me this is an asymptotic analysis. You need to be sufficiently close to the optimum so that the bound can kick in (i.e., its leading coefficient to be smaller than 1). It would seem a good idea to present line 199-207 in greater detail since this is a strong example. (But to note that the authors needed Lipschitz continuity of the second derivative, which is unusual.) I do not agree with the authors in "A fair comparison, since the number of gradient oracle calls was identical in both cases and the grid search on lambda had only a marginal impact on complexity." Note that Nesterov backtracking, if implemented correctly, does not require additional gradient calls, very much the same as the authors backtracking on lambda. All experiments in this work are reported with x-axis being the number of gradient oracle calls. Another figure with x-axis the cumulative running time should also be presented, as RMPE has some overheads (solving Eq. (17) etc.). ===== end ===== I very much like how the authors relate classical extrapolation methods to the more recent accelerated gradient methods (e.g., Nesterov's). No doubt this will provide new angles to understand acceleration. However, the presentation of this work needs some improvement, and the main results need to be better articulated and contrasted. The experiments are also week for reasons I will detail below. Let me first comment that the approximate MPE method is NOT novel, in contrast to what the authors claimed in line 58. It is interesting that the authors cited [19] multiple times but did not appear to recall its main points. For instance, (APME) below line 98 is exactly Eq. (10.5) of [19], which attributed to Mesina. Also, the bound above line 102 is exactly Eq. (10.6) of [19]. From this point of view, claiming novelty of APME, or even renaming it as APME, is not appropriate. Another example: Theorem 1 in [19] did not prove the claim in footnote 2 at all (in fact the claim is not true: one needs q to be minimal). This kind of sloppiness is not unusual in the whole manuscript, which is hurting the credibility of this work. Other examples: Eq. (5) used U_0, which is never defined, and then in Eq. (5) and afterwards, it was changed to u_0; The derivations in Eqs. (6)-(9) should have the constraint deg(p) <= k; etc. I would strongly suggest the authors spend more time on explaining their results, and better contrast with existing methods. For instance, it should be mentioned that under the condition (on A) in Prop. 2.2, the original sequence x_i is in fact converging at a linear rate. Shouldn't you care to mention this and compare the constants (e.g., slope of linear rate)? The discussions from line 119 to line 129 are very interesting, but they lack some details and references. These comparisons, to me, are the interesting part, and I wish the authors to be more elaborative and explicit. The regularized AMPE in section 24 is not well-motivated. First, one should point out that all the analyses are asymptotic, namely that the approximation or expansion is only valid when x_i is sufficiently close to x_*. The asymptotic nature and the many implicit constants of this analysis make the justification of RAMPE shaky. Second, the authors' motivation to propose RAMPE is to avoid any near-singularity of U^T U. But, when U^T U is near singular, it clearly suggests that the principle behind AMPE may well be violated (e.g., x_i not converging linearly, k too small, etc.). In that case, it seems more useful to fix AMPE as suggested in the reference [19], instead of regularizing? My biggest complaint comes from the experiment, in particular the selection of the parameter lambda. The authors used a backtracking strategy, which is fine but directly contradicts what is said in line 178, since multiple evaluation of the function f is indeed needed. In this regard, the experiment is not fair either: First, one should also compare the cumulative running time (on top of gradient evaluations), since Nesterov's acceleration does not cost any computational overhead while the proposed RAMPE does. Second, one should also allow Nesterov's algorithm to backtrack the Lipschitz constant and strong convexity parameter, like in [3, 4]. I suspect the main difference in Figs 1, 2 are because of this backtracking (of lambda). The authors need to convince me otherwise. Also, is the condition A < I satisfied in the experiments? (I suspect not since the condition number was deliberately set too large, e.g. tau = 10^2.)

Confidence in this Review

2-Confident (read it all; understood it all reasonably well)


Reviewer 3

Summary

This paper proposes approximate minimal polynomial extrapolation (AMPE) and its regularized version (RMPE), which are modifications of minimal polynomial extrapolation (MPE) and are claimed to accelerate convergence. Errors bounds and experimental comparisons were provided.

Qualitative Assessment

I am unclear about the following aspects: 1. What is p^*(A) in (6) and why the first inequality holds? Note that p(x) is not chosen as the Chebeshev's polynomial. Its coeffients are simply a constrained least regression solution. 2. What is NestrovConv in Figures 1&2? Why it sometimes appears in the comparison (Fig. 1) and sometimes not (Fig. 2) (Also see the supplementary material)? 3. How to compute f(x^*) in Figures 1&2? Finally, it is quite annoying that some reference numbers are missing. *************************** Comments after reading the author feedback: The authors' feedback only addressed part of my concerns. As for the most ciritical comment: whether C can be smaller than 1, the authors actually did not have a solid proof. They just referred me to Prop. 2.6 (which I doubted) and "fig A.3" (which I did not find in either the paper or the supplementary material. I guess that it is Fig. 3 in the supplementary material). For whether the C can be smaller than 1, the analysis in the middle of page 7 is only heuristic. The second parameter, \lambda/||x_0 - x^*||^2, in S cannot be 0 anyway (If \lambda = 0, in Prop. 2.6 the other factor before S will be infinity). Figure 3 in supplementary material shows that this parameter increases when k increases, although indeed very small. In short, the empirical evaluations are in favor of the proposed exrapolation technique, but the theoretical analysis did not convince me. The paper seemed to be done in a haste (The authors blamed Dropbox, which is strange to me. Are they collaborating via dropbox?).

Confidence in this Review

3-Expert (read the paper in detail, know the area, quite certain of my opinion)


Reviewer 4

Summary

In numerical analysis, there is a well known technique called series acceleration. These techniques provide a collection of sequence transformations for improving the rate of convergence of a series. In this paper, they show how to apply these series acceleration on convex optimization.

Qualitative Assessment

I find this idea of applying series acceleration in optimization is very interesting. Although this paper is only able to recover the Nesterov acceleration for the linear case and some limited result for non-linear case, I think this is a good start. I believe it is likely that there are other application of series acceleration in optimization and it is a very good opportunity for ML people in general to learn about this. Therefore, I highly recommend this paper.

Confidence in this Review

3-Expert (read the paper in detail, know the area, quite certain of my opinion)


Reviewer 5

Summary

The paper presents a method of employing the history of iterates from an optimization algorithm in order to produce better estimates, that can be heuristically used in order to produce speedups. The intuition relies on the fact that the Chebyshev iteration relies on constructing a polynomial of low degree which evaluates to close to 0 when applied to the eigenvalues of the input matrix. This intuition is extended into defining the minimal polynomial of a matrix, which fulfills exactly that purpose, and computing a generalization of it during each iteration of the algorithm. A similar method is used in the nonlinear case.

Qualitative Assessment

The ideas in this paper are very interesting, but the writeup is somewhat confusing. Specifically, I don’t understand whether the convergence bounds from section 2.5 only apply to (16), or if they can be extended to more general settings. Also, a more elaborate discussion of why the minimal polynomial is the right object to consider would be in order in section 2.1.

Confidence in this Review

1-Less confident (might not have understood significant parts)


Reviewer 6

Summary

In this paper, authors analyse approximate minimal polynomial extrapolation scheme. One can run e.g. gradient algorithm to produce iterates \{x_i\} and then AMPE can provide an estimate of solution (fixed point).

Qualitative Assessment

I like the main idea of the paper. However, I do not see it to be very practical. The numerical experiments suggest it could be working fine, but I can see there some issues. To convince a reader it would be nicer to see also another maybe more standard datasets in experiments. Also, I am not sure if the choice of \tau=100 in numerical experiments is somehow too large, isn't it? Also I do not see how L/tau is 10^9? Have you normalized \xi_i such that \|\xi_i\| \leq 1 or not? I think that the general idea is nice and can influence others, but should be developed further. The issues I can see with the paper are as follows: 1. the linear model it realistic e.g. for quadratic function, but I take it that it was good to start with it. However, once you have a non-linear model things gets more complicated. I would really like to see the section 2.3. to be more detailed as it is the most important one (otherwise, I can just use CG). 2. it was somehow hard for me to follow the paper. I was not sure what is assumed to hold in various portion of the paper, e.g. Section 2.4 assumes (3) or (12) to hold? Of course after thinking for some time, I get some understanding of what roughly it says, but the presentation should be made such that it is immediately clear. 3. Also the paper refers to Nesterov method. To be honest, Nesterov has so many methods. Also in Figure 1, there are two Nesterov algorithms which one is which? Also which line corresponds to nonlinear CG method? 4. If I am not mistaken, the sensible theoretical guarantee is provided only for linear model, or an intuition is provided when x_0 is close to x^*, which is somehow still ok, but maybe the theory could be better developed and presented.

Confidence in this Review

2-Confident (read it all; understood it all reasonably well)